# Association between Facebook Addiction, Depression, and Emotional Regulation among Women

**DOI:** 10.3390/healthcare11121701

**Published:** 2023-06-09

**Authors:** Atallah Alenezi, Warda Hamed, Ibrahim Elhehe, Rania El-Etreby

**Affiliations:** 1College of Applied Medical Sciences, Shaqra University, Shaqra 11961, Saudi Arabia; dribrahim1987@mans.edu.eg; 2Faculty of Nursing, Mansoura University, Mansoura 35516, Egypt or hamed.warda@yahoo.com (W.H.); rania.etreby@yahoo.com (R.E.-E.); 3College of Nursing, Jazan University, Jazan 82812, Saudi Arabia

**Keywords:** depression, emotional regulation, Facebook addiction

## Abstract

Facebook has grown to be one of the most widely used communication platforms. A new condition linked with widespread use has emerged with its expanding popularity: Facebook addiction. A descriptive cross-sectional research design was used in the current study, in two randomly selected villages (Elmanial and Batra) and one town (Talkha city) affiliated with Dakahlia Governorate, Egypt. Data were collected from women only through using a self-administered socio-demographic questionnaire, Facebook addiction scale, Beck depression inventory-II, and emotional regulation scale. The study found 83.7% of the studied women reported moderate emotional regulation; 27.9% had moderate Facebook addiction, and 23.9% reported mild depression. The study concluded there was a significant negative correlation between Facebook addiction and emotional regulation.

## 1. Introduction

Facebook has grown to become one of the most common social media platforms. It has been in operation since 2004 [1]. This internet platform may be used for entertainment, communication, building relationships, self-promotion, and marketing. People spend a lot of time on Facebook, as evidenced by the statistics [2]. There have been some interesting studies into social media addiction over the last few years. Most of these studies have concentrated on “Facebook Addiction”, which has been categorized as a probable behavioral addiction by some researchers and is considered one of the most general internet addiction research areas [1,3,4].

Addiction is defined as the inability to stop using drugs despite previous attempts [5]. Addiction to social networking sites has recently attracted the attention of academics. Excessive worry about social media sites or a strong desire to log on to social media sites was described as interfering with normal social activities and/or mental health and well-being [6]. Obsessively monitoring social media behavior has been investigated in terms of usage patterns, motivations, and negative repercussions. These findings show that, in addition to excessive usage of social media sites leading to addiction, the young generation’s addiction to social media may be the result of intra-psychic conflict [7]. According to studies, social networking sites have detrimental effects on mental health, such as anxiety, sleeplessness, and community loss in real-life settings [8].

Facebook addiction is a type of non-chemical addiction that involves excessive engagement in human–machine communication. It fulfills six fundamental addiction specifications, which are salience, mood adjustment, tolerance, withdrawal symptoms, conflict, and relapse [9]. The use of social media platforms, such as Facebook, has been associated with various socio-demographic aspects such as age, gender, marital status, and professional/educational background, which can influence patterns of Facebook usage [10,11]. Participating in fewer daily physical activities, drug and alcohol dependency, greater time spent on Facebook, loneliness, poor sleep quality, and feeling unhappiness in a relationship have all been identified as predictors of problematic Facebook usages [12,13]. Numerous studies have demonstrated a strong association between excessive Facebook use and negative impacts on mental health, as well as the development of physical and psychological issues that can adversely affect psychological well-being. Specifically, research has linked excessive Facebook use to increased rates of depression [12,14]. Facebook depression occurs when persons consume a considerable and extreme amount of time on Facebook and starts to show typical symptoms of depression [15].

Emotion regulation is described as the process through which humans consciously and unconsciously modify their emotions to respond correctly to environmental demands [16]. Furthermore, emotional regulation has been linked to psychological distress, as some disorders (such as depression and anxiety) might be linked to a lack of emotional control. Likewise, several studies have also recognized psychological distress as being associated with Facebook addiction [17,18]. Addiction and emotion regulation have been proven to be linked [19]. Addictions are associated with a lack of emotional control [20]. When it comes to emotion regulation, suppression among emotion regulation strategies is not restricted to emotions; suppressing thoughts is also a way of dealing with addictions. To control their desires, individuals with addictions strive to hide their thoughts. Hormes et al., reported that those with high degrees of Internet addiction had difficulty controlling their emotions [21].

People with Facebook addiction and the escalation of the severity of their addiction may affect their physical and mental health and performance of their social roles. Therefore, psychiatric nurses play an important role in preventing Facebook addiction in women, especially those with a high addiction risk. It is suggested that nurses cooperate with women to provide them with proper education about contributing factors to Facebook addiction.

### 1.1. Significance of the Study

Excessive and uncontrolled use of social media (especially Facebook) has lately gotten a lot of interest, yet organizations like the American Psychiatric Association and the World Health Organization still do not diagnose it as a recognized mental health disorder [11,18]. However, there is compelling evidence that social media use (particularly Facebook use) is harmful to many individuals and that it shares many characteristics with behavioral addictions like gambling and gaming [14]. Globally, there are 2.80 billion Facebook users; 37.5% of them are women. Furthermore, more than 350 million people globally are deemed to satisfy the clinical criteria of addiction [22]. Studies have shown that Facebook addiction activates the same brain areas as substance addiction [23]. Few studies that have been conducted in Arabic exist showing the relationship between Facebook addiction in women, their depression, and their emotional regulation. This study will provide a guideline for focusing on the contributing factors to depression in women, such as Facebook addiction and emotional regulation. Early detection of levels of Facebook addiction and degrees of emotional regulation will open a new approach in the nursing management of depressed women. The objective of the research was to evaluate the prevalence of Facebook addiction, depressive symptoms, and emotional regulation among women and investigate the relationship between Facebook addiction, depressive symptoms, and emotional regulation in the female population.

### 1.2. Research Questions

The study aims to explore the following research questions:Is there a significant relationship between Facebook addiction and depressive symptoms among women?Is there a significant relationship between Facebook addiction and emotional regulation among women?Is there a significant relationship between depressive symptoms and emotional regulation among women?

## 2. Methods

### 2.1. Study Design/Setting and Subjects

This study used a descriptive cross-sectional research design. The study was carried out in two randomly selected villages (Elmanial and Batra) and one town (Talkha City) affiliated with Dakahlia Governorate, Egypt. All the studied women met the following inclusion criteria: using Facebook; being free from physical, mental, and cognitive disease; having no history of mental illness; and agreeing to participate in the study.

The sample size:

Sample size of the current study calculated based on the following equation.

n = [(Z _1−α/2_)^2^ ∗ P (1 − P)]/d^2^ considering precision/absolute error of 5% and type I error of 5%, where Z _1−α/2_ = is the standard normal variety. At 5% type I error (*p* < 0.05), it is 1.96. P = the expected proportion in population based on previous studies [24]. D = absolute error or precision. So, sample size = [(1.96)^2^(0.379) (1 − 0.379)]/(0.06)^2^ = 251 participants.

### 2.2. Tools of Data Collection

The study utilized an online self-administered questionnaire, including the following tools:

Tool I: Socio-Demographic Data Questionnaire: The researchers developed this questionnaire to gather data on the participants’ age, income, educational level, marital status, occupation, and income.

Tool II: The Facebook Addiction Scale was adopted and modified from the Bergen Facebook Addiction Scale by Andreassen et al. [25]. Originally, the Bergen Facebook Addiction Scale consisted of eighteen items, three for each of the six basic components of addiction (salience, mood modification, tolerance, withdrawal, conflict, and relapse). Each item is graded on a 5-point scale with values of 1: very rarely and 5: very often. The sum of scale items ranged from 18 to 90. Higher scores indicate greater Facebook addiction. Total scores were categorized into three levels: low, moderate, and high. Cronbach’s alpha was 0.83 in this study. Previous study reported that the reliability test of the scale using Cronbach’s alpha was 0.82 [13].

Tool III: Beck Depression Inventory-II (BDI-II): This scale was established by Beck et al. [26] and is a widely used self-reported measure of depression. The BDI-II consists of 21 items, with each item rated on a scale ranging from 0 to 3. Higher scores indicate greater levels of depression. A score of 0–13 indicates no depression; a score of 14–19 indicates mild depression; a score of 20–28 indicates moderate depression, and a score of 29–63 indicates severe depression. Cronbach’s alpha was 0.79 in this study. Previous study reported that the reliability test of the scale using Cronbach’s alpha was 0.9 [27].

Tool IV: Emotional Regulation Scale: This scale was designed to measure the ability to regulate emotions [28]. The scale includes a series of questions regarding an individual’s emotional life and how he or she manages (or regulates) his or her emotions. The questions are about two different parts of a person’s emotional existence. The first is his emotional reaction, or how he feels on the inside. The other is his emotional expression, or how he expresses his feelings through his words, gestures, and actions. The scale consists of 10 items under two dimensions: reappraisal (6 items) and suppression (4 items). Each item was rated on a 7-point Likert scale ranging from 1 (strongly disagree) to 7 (strongly agree). Higher scores indicate greater emotional regulation. Total scores were categorized into three levels: low, moderate, and high. Cronbach’s alpha was 0.81 in this study. Previous study reported that the reliability test using Cronbach’s alpha was 0.79–0.88 for subscales [29].

### 2.3. Ethical Consideration

The approval of the Research Ethics Committee, Faculty of Nursing, Mansoura University, was obtained (Ref.No.P.0381). After the aim of the research was described, subjects gave their informed consent to participate in the study voluntarily. They have the right to refuse participation and withdraw from the research at any time and without reason. They were also assured that their data would be kept confidential and used only for research purposes.

### 2.4. Pilot Study

A pilot study was carried out on 20 women who were purposefully selected from both rural and urban areas to ensure diversity in our sample, assess the feasibility of our study, and test the tools that we will be using for the larger study. Specifically, we wanted to identify any errors or issues with the instruments and to fine-tune them before proceeding with the master data collection. 

### 2.5. Field of Work

An approval letter was issued from the directors of the social clubs in the selected settings to carry out the study. In order to ensure the accuracy and dependability of construct measurement across various cultural and linguistic contexts, the researchers translated data collection tools into Arabic and then back translated them. Additionally, this process ensures that the scale is culturally appropriate and pertinent to the intended participants. It was also tested for its content validity by a jury of five experts in the psychiatric and mental health nursing fields, and the necessary modifications were made. The tool’s reliability was tested through Cronbach’s alpha test.

The data collection period started in December 2021 and ended in March 2022. The researchers met the participants at social clubs which are available in rural and urban areas, (it contains an entertainment places for playing, sitting, and having meals and drinks). They provided participants with an explanation of the study’s importance, goal, and benefits of participating. To conduct the study, the researchers employed an online Google Forms spreadsheet. They shared a link with the participant women to collect data that included an online questionnaire. The link was shared through What’s App groups. The researchers used the What’s App group to explain to the women the components of the tools and how to answer the online questionnaire. The time needed to complete the study tools ranged from 20 to 30 min.

### 2.6. Statistical Design

The collected data were organized, tabulated, and statistically analyzed using SPSS software (Statistical Package for the Social Sciences, version 22, SPSS Inc., Chicago, IL, USA). The categorical variables were represented as frequency and percentage. An independent t-test was used to test the difference between two means of continuous variables. An ANOVA test was used to test the difference between more than two means of continuous variables. The Pearson correlation coefficient test was conducted to test the association between two continuous variables. Regression analysis was conducted to explore independent variables of Facebook addiction. Statistically significant was considered as (*p*-value ≤ 0.01 and 0.05).

## 3. Results

### 3.1. Socio-Demographic Characteristics of the Studied Women

Table 1 illustrates how the women in the study were distributed based on their socio-demographic factors. This table shows that 58.6% of the women studied were between the ages of 22 and 35 and that more than half of them resided in rural areas, were married, and had higher education degrees (54.2, 89.2%, and 51.8%, respectively). In addition, 52.2% of them were not working, and 85.7% reported enough income. The most of the studied women did not have diseases (81.3%).

### 3.2. Prevalence of Facebook Addiction, Depressive Symptoms, and Emotional Regulation among the Studied Women

Table 2 displays the distribution of the studied women according to their levels of Facebook addiction. It was shown that 45.0% had no Facebook addiction, 27.9% had moderate Facebook addiction, and only 6.0% had a high level of Facebook addiction.

Table 3 reveals levels of depression among the studied women. It was observed that 41.4% of the women reported no depression, while mild and moderate depression was reported by 23.9% and 18.7%, respectively.

Table 4 shows levels of emotional regulation among the studied women. It was observed that 83.7% of the women reported moderate emotional regulation. More than half reported moderate reappraisal (61.0%), while 12.4% reported high suppression.

### 3.3. Differences of Facebook Addiction, Depression, and Emotional Regulation in Relation to Socio-Demographic Characteristics of the Studied Women

Table 5 illustrates the relationship between Facebook addiction, depression, emotional regulation, and socio-demographic characteristics of the studied women. It appears from the table that there is a significant relationship between marital status and depression and a significant relationship between emotional regulation and monthly income (*p* = 0.04, *p* = 0.04, respectively) but no significant relationship between Facebook addiction and the studied women’s socio-demographic characteristics.

### 3.4. Relationship between Facebook Addiction, Depression, and Emotional Regulation in Relation to Socio-Demzographic Characteristics of the Studied Women

Table 6 describes the correlation between emotional regulation, Facebook addiction, and depression among the studied women. It was observed that there is a significant, negative correlation between Facebook addiction and emotional regulation (−0.42, *p* = 0.000). There was no correlation between depression and Facebook addiction (*p* = 0.99).

Table 7 shows that emotional regulation has a negative unstandardized coefficient (B = −0.75) which indicates that an increase in emotional regulation is associated with a decrease in Facebook addiction. There is a moderate negative relationship between emotional regulation and Facebook addiction (β = −0.42). Emotional regulation is a significant negative predictor of Facebook addiction.

Table 7 shows that emotional regulation has a negative unstandardized coefficient (B = −0.75) which indicates that an increase in emotional regulation is associated with a decrease in Facebook addiction. There is a moderate negative relationship between emotional regulation and Facebook addiction (β = −0.42). Emotional regulation is a significant negative predictor of Facebook addiction.

## 4. Discussion

Disordered Facebook use appears to be part of a symptom set that includes a lack of emotional management skills and an increased risk of substance and non-material addictions [21]. Facebook is one of the most widely used social media and communication platforms. However, the various benefits of utilizing this service, it may also lead to addiction in certain cases [21]. Therefore, the objective of the research was to evaluate the prevalence of Facebook addiction, depressive symptoms, and emotional regulation among women and investigate the relationship between Facebook addiction, depressive symptoms, and emotional regulation in the female population.

### 4.1. Prevalence of Facebook Addiction, Depressive Symptoms, and Emotional Regulation among Women

The current study results indicate that the study sample has various levels of Facebook addiction. Only about a third of participants reported moderate degrees of Facebook addiction, and only about a tenth of them had severe levels of Facebook addiction. However, nearly half of the study participants did not have a Facebook addiction. Furthermore, the results refute what Cudo claimed that females are more vulnerable to Facebook addiction and gender is a predictor of Facebook addiction. He also claimed that women had a stronger correlation between Facebook usage and Internet addiction than men [30]. Moreover, this result could be attributed to the busy schedules of the women, as nearly half of the study sample are working and living in rural areas. This explanation is in line with what Khorazaty states, that more and more women are also taking on roles that were originally male ones or assuming sole responsibility for agricultural production and livestock farming [31].

Furthermore, the current data contradict Luarn’s claim that women use Facebook more for liking, commenting, and messaging [32]. Furthermore, the results refute Cudo’s claim that females are more vulnerable to Facebook addiction and that gender is a predictor of Facebook addiction. He also claimed that women had a stronger correlation between Facebook usage and Internet addiction than men [30]. 

### 4.2. Facebook Addiction, Depression, and Emotional Regulation in Relation to Socio-Demographic Characteristics of the Studied Women

The present study revealed that there was no significant relationship between Facebook addiction and most socio-demographic data. This disagrees with a study made by Sotero et al., which stated that higher Facebook use leads to a higher risk of addiction in women aged between 24 and 30 [33]. Current study results are regarded to the similarity of the subjects in their socio-economic characteristics, as most of them range in the same age group (22–35), their residency in the agricultural governorate, and the same customs, traditions, social relations, and interests. Hence, that made no difference in most of the mean scores of Facebook addiction in the sample regarding their socio-demographic data.

For emotion regulation, the current study revealed that about three quarters of the study sample have a moderate level of emotion regulation. Specifically, more than two-thirds of them are moderately reappraising their emotions. This result agrees with a study by Chaplin and Aldao, who stated that girls employ more adaptive emotion regulation methods, such as re-evaluation of active coping, and less maladaptive strategies, such as rumination and suppression [34]. Furthermore, according to McRae et al., women can employ good emotions to reappraise bad emotions to a greater extent [35]. As they live in an agricultural governorate, this might be tied to their habits and traditions. That corresponds with Kang’s findings that culture influences how people interpret events in their lives and influences how they express their emotions [36]. As a result, understanding one’s own and others’ cultures will assist in preventing disputes and controlling one’s emotions.

The study findings showed that there is a significant relationship between emotion reappraisal and income, specifically women who had insufficient income demonstrated higher levels of emotion reappraisal. This result is consistent with previous research by Hittner et al. which found that socioeconomic position strongly influences concurrent and longitudinal emotion reappraisal. Specifically, individuals from low-socioeconomic backgrounds tend to benefit the most from reappraisal, while those from high-socioeconomic backgrounds benefit the least [36]. These results have important implications for understanding the role of income in emotional regulation and suggest the need for further research on the impact of socioeconomic factors on mental health outcomes [37].

For depression, this study revealed that more than one-third of the sample had no depression, and one quadrant had mild to moderate depression. Additionally, nearly one-third of the study subjects had moderate to severe depression. That could be related to a decreased number of individuals with Facebook addiction as nearly two-fifths of the sample have no Facebook addiction and about one-fifth has a low level of Facebook addiction. This agrees with Al Mamun and Griffiths who stated that Facebook addiction was a predictor of depression in Bangladeshi students [3].

### 4.3. Relationship between Facebook Addiction, Depression, and Emotional Regulation among the Studied Women

There is no association between reappraising of emotion and levels of depression in this study, and this disagrees with what Orgilés et al. said that adaptive strategies are weakly related to depressive symptoms, while maladaptive strategies are strongly related to depressive symptoms [38].

Moreover, the results disagree with research made by Wu et al., who stated that emotion dis-regulation has long been thought to be a vulnerability factor for mood disorders. He also added that women with depression have higher reported spontaneous emotion suppression levels and lower levels of reported emotion reappraisal [39].

The presence of depression could be regarded as the correlation of depression with Widowhood, and this agrees with Sasson and Umberson who stated that widowhood is linked to greater financial burden, additional household management responsibilities, and changes in social interactions, all of which can worsen or alleviate psychological distress and depression [40].

The present study also reported that there is no relationship between Facebook addiction and depression among women. One reason for this finding is that women may use Facebook for better social contact and emotional support, which may offset the detrimental consequences of excessive use on their mental health. Another possibility is that women have distinct coping mechanisms for dealing with stress and negative emotions, which may be more efficient in minimizing the harmful consequences of Facebook addiction on their mental health. This finding disagreed with the other researchers who reported a significant positive correlation between Facebook addiction and depression among participants [3,27].

Finally, the results revealed that Facebook addiction is significantly affected by emotional regulation. That could be related to the positive emotion regulation of most of the study sample as nearly two-thirds of them have moderate emotional reappraisal. That result agrees with Caplan who stated that there was a negative relationship between Facebook addiction and functional emotion regulation [41]. It also agrees with a study by Mehdizadeh who revealed that persons with low self-esteem and low emotion regulation had a greater number of Facebook logins per 12 days [42]. Hence, Facebook usage may be a type of emotional regulation. Ozimek et al. support this concept, stating that individuals may use Facebook to change their mood, which could be a useful tool and an indication of Facebook addiction [43].

### 4.4. The Implications of the Study Findings

The study’s findings suggest that mental health interventions aimed at enhancing emotional regulation skills may be effective in reducing Facebook addiction among women. The aforementioned study highlights the need for additional research on the frequency and repercussions of Facebook addiction and associated mental health concerns among women. Our study examined Facebook addiction in women, but the addictive qualities of social media and its potential effects on mental health and emotional regulation may also be relevant to young people who frequently use social media. The present study contributes to the increasing literature emphasizing the necessity of preventive education regarding the hazards of social media usage, particularly among the youth. Further research is needed to investigate the efficacy of educational interventions in mitigating social media addiction and its adverse consequences among young people. Psychiatric nurses have an important role in educating their patients about the impact of excessive face book use on their emotional regulation and their levels of depression.

### 4.5. Limitations

This study has several limitations that should be considered. First, the sample was limited to women, which may limit the generalizability of the results to other populations, such as men. Second, the study was conducted within a specific cultural and social context which may limit the findings’ applicability to other cultures and contexts. The study also did not examine the potential influence of other variables, such as social support or coping strategies, on the relationship between Facebook addiction, depressive symptoms, and emotional regulation. These limitations may have an effect on the external validity of the study’s findings.

## 5. Conclusions

The study concluded that nearly half of the participants did not exhibit Facebook addiction, while only one-third had a moderate-to-severe addiction. The majority of the women in the study demonstrated moderate emotional regulation, and over half reported moderate reappraisal. Two-fifths of the sample showed no symptoms of depression, while less than one-fifth had severe depression. The study also revealed a statistically significant inverse correlation between Facebook addiction and emotional regulation in females. Specifically, the findings suggest that Facebook addiction increases as emotional regulation decreases. These results highlight the importance of emotional regulation in the development and perpetuation of Facebook addiction among women.

## Figures and Tables

**Table 1 healthcare-11-01701-t001:** Distribution of the studied women according to their socio-demographic characteristics (*n* = 251).

Characteristic	*n* (%)
Age years:	
18–21	10 (4.0)
22–35	147 (58.6)
>35	94 (37.5)
Residence	
Urban	115 (45.8)
Rural	136 (54.2)
Marital status	
Single	17 (6.8)
Married	224 (89.2)
Divorced	7 (2.8)
Widowed	3 (1.2)
Level of education	
Diploma degree	41 (16.3)
Secondary degree	16 (6.4)
Higher education degree	130 (51.8)
Postgraduate studies	64 (25.5)
Occupation	
Working	120 (47.8)
Not working	131 (52.2)
Income	
Enough	215 (85.7)
Not enough	36 (14.3)
Having diseaes	
Yes	47 (18.7)
No	204 (81.3)

**Table 2 healthcare-11-01701-t002:** Levels of Facebook addiction among the studied women.

Levels Facebook Addiction	Score	*n* (%)
No addiction	18–35	113 (450)
Low	36–44	53 (21.1)
Moderate	45–67	70 (27.9)
High	68–90	15 (6.0)

**Table 3 healthcare-11-01701-t003:** Levels of depression among the studied women.

Degrees of Depression	Score	n (%)
No depression	0–13	104 (41.4)
Mild to moderate depression	14–19	60 (23.9)
Moderate to severe depression	20–28	47 (18.7)
Severe depression	29–63	40 (15.9)

**Table 4 healthcare-11-01701-t004:** Levels of emotional regulation among the studied women.

Emotional Regulation	Levels of Emotional Regulation	Scores	*n* (%)
Emotional regulation	Low (<50%)	10–34	15 (6.0)
	Moderate (50–75%)	35–52	210 (83.7)
	High (>75%)	53–70	26 (10.4)
Reappraisal	Low (<50%)	6–20	32 (12.7)
	Moderate (50–75%)	21–31	153 (61.0)
	High (>75%)	32–42	66 (26.3)
Suppression	Low (<50%)	4–13	81 (32.3)
	Moderate (50–75%)	14–21	139 (55.4)
	High (>75%)	22–28	31 (12.4)

**Table 5 healthcare-11-01701-t005:** Mean scores differences of Facebook addiction, depression, and emotional regulation in relation to socio-demographic characteristics of the studied women.

Characteristics	Facebook Addiction	Depression	Emotional Regulation
Age years:	Mean ± SD	Mean ± SD	Mean ± SD
18–21	31.40 ± 12.81	17.30 ± 9.84	42.40 ± 7.31
22–35	40.51 ± 16.39	17.42 ± 11.26	42.98 ± 7.99
>35	38.17 ± 15.31	18.23 ± 11.48	45.26 ± 10.11
F value/*p*	1.90/0.15	0.16/0.86	2.04/0.13
Residence			
Urban	40.99 ± 16.08	16.93 ± 11.30	43.54 ± 9.07
Rural	37.24 ± 15.59	18.65 ± 11.18	44.13 ± 8.34
t-value/*p*	1.86/0.06	1.21/0.23	0.53/0.60
Marital status			
Single	42.71 ± 14.50	13.94 ± 13.52	43.06 ± 7.60
Married	39.00 ± 15.99	17.68 ± 10.90	43.65 ± 8.96
Divorced	35.14 ± 16.51	21.43 ± 13.76	51.14 ± 7.78
Widowed	49.67 ± 20.21	33.33 ± 5.51	42.67 ± 3.06
F value/*p*	0.87/0.46	2.88/0.04 *	1.69/0.17
Level of education			
Diploma degree	35.27 ± 14.37	20.10 ± 10.21	45.41 ± 7.22
Secondary degree	35.94 ± 15.90	15.50 ± 9.17	45.13 ± 9.99
Higher education degree	39.89 ± 16.22	17.47 ± 11.65	43.31 ± 8.72
Postgraduate studies	41.41 ± 16.09	17.27 ± 11.55	43.47 ± 9.80
F value/*p*	1.56/0.20	0.87/0.46	0.73/0.53
Occupation			
Working	40.22 ± 16.60	16.68 ± 11.48	42.99 ± 9.26
Not working	38.40 ± 15.32	18.68 ± 11.00	44.56 ± 8.45
t-value/*p*	0.90/0.37	1.41/0.16	1.40/0.16
Monthly income			
Not enough	39.08 ± 17.65	20.25 ± 11.32	46.58 ± 10.46
Enough	39.30 ± 15.68	17.30 ± 11.22	43.34 ± 8.50
t-value/*p*	0.08/0.94	1.46/0.15	2.04/0.04 *
Having disease			
Yes	38.79 ± 14.36	16.81 ± 10.43	43.49 ± 9.88
No	39.38 ± 16.31	17.93 ± 11.45	43.88 ± 8.63
t value/*p*-value	0.23/0.82	0.62/0.54	0.27/0.79

* Statistically significant (*p* ≤ 0.05).

**Table 6 healthcare-11-01701-t006:** Relationship between emotional regulation, Facebook addiction, and depression among the studied women.

	Depression		Facebook Addiction	
	r	*p*	r	*p*
Emotional regulation	0.11	0.08	−0.42	0.000 **
Facebook addiction	0.001	0.99	1	
Depression	1		0.001	0.99

** highly statistically significant (*p* ≤ 0.01).

**Table 7 healthcare-11-01701-t007:** Effect of emotional regulation on Facebook addiction among the studied women.

Independent Variable	Unstandardized Coefficients		Standardized Coefficients	T	Sig.
	B	Std. Error	β		
Emotional regulation	−0.75	0.10	−0.42	7.25	0.000 **

** highly statistically significant (*p* ≤ 0.01).

## Data Availability

The datasets generated and/or analyzed in the current study are available from the corresponding author upon request. The data are not publicly available due to privacy.

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
