# Peer review of "Association between Facebook Addiction, Depression, and Emotional Regulation among Women"

_healthcare, 2023, doi:10.3390/healthcare11121701_

Round 1

Reviewer 1 Report

Thank you for the opportunity to review this study and article.  I believe with some small amendments and additions this will be appropriate for publication.

Lines 47-49  Reads awkwardly.  Needs minor adjustments.

Line 85- reword.  Little to no research in Arab nations exists.

Line 86- citation needed for quote

Line 88 and 89 Facebook

Line 122 Facebook

Line 158 Write a little more here as to your purpose of the pilot study, your recruitment of them and IRB.  If these 20 subjects were randomly pulled from your full 200+ population, note that for clarity.

256- check use of apostrophe

274 girls

284 concurrent

290 decreased

316 researchers who reported

I believe you can also make a call that your study adds to the need for preventative education with youth on the dangers of social media.  While this study focuses on Facebook which is now more focused and used by adults, your findings may parallel those about social media challenges for youth.  Add this to your discussion.

Same as above:

Thank you for the opportunity to review this study and article.  I believe with some small amendments and additions this will be appropriate for publication.

Lines 47-49  Reads awkwardly.  Needs minor adjustments.

Line 85- reword.  Little to no research in Arab nations exists.

Line 86- citation needed for quote

Line 88 and 89 Facebook

Line 122 Facebook

Line 158 Write a little more here as to your purpose of the pilot study, your recruitment of them and IRB.  If these 20 subjects were randomly pulled from your full 200+ population, note that for clarity.

256- check use of apostrophe

274 girls

284 concurrent

290 decreased

316 researchers who reported

I believe you can also make a call that your study adds to the need for preventative education with youth on the dangers of social media.  While this study focuses on Facebook which is now more focused and used by adults, your findings may parallel those about social media challenges for youth.  Add this to your discussion.

Author Response

Dear reviewer

Thank you for taking the time to review our study and for providing valuable feedback. We hope that our revisions have addressed your concerns, and we look forward to hearing back from you for further considerations if needed.

Point 1: Lines 47-49  

Response 1: Adjustments done in line 45-50

Point 2: Line 85- reword.  Little to no research in Arab nations exists.

Response 2:   modification done in line 87

Point 3: Line 86- citation needed for quote

Response 3: English editing done

Point 4: Line 88 and 89 Facebook

Response 4: modification done in line 90,91

Point 5: Line 122 Facebook

Response 5: modification done in line 123

Point 6: Line 158 Write a little more here as to your purpose of the pilot study, your recruitment of them and IRB.  If these 20 subjects were randomly pulled from your full 200+ population, note that for clarity.

Response 6: modification done in line 153, 165-170

Point 7:  256- check use of apostrophe

Response 7: modification done in line 264

Point 8: 274 girls

Response 8: modification done in line 285

Point 9: 284 concurrent

Response 9: modification done in line 300

Point 10: 290 decreased

Response 10: modification done in line 310

Point 11: 316 researchers who reported

Response 11: modification done in line 335

Point 12: I believe you can also make a call that your study adds to the need for preventative education with youth on the dangers of social media.  While this study focuses on Facebook, which is now more focused and used by adults, your findings may parallel those about social media challenges for youth.  Add this to your discussion.

Response 12: response added in line 353-359

Our study examined Facebook addiction in women, but the addictive qualities of social media and its potential effects on mental health and emotional regulation may also be relevant to young people who frequently use social media. The present study contributes to the increasing literature emphasizing the necessity of preventive education regarding the hazards of social media usage, particularly among the youth. Further research is needed to investigate the efficacy of educational interventions in mitigating social media addiction and its adverse consequences among young people.

Sincerely

Reviewer 2 Report

Hi

I found the English expression to be poor and I was distracted by it while reading the article.

Try not to speak in the passive voice. So instead of writing "In a study conducted by Hormes et al., it was discovered that ..." write: " Hormes et al., reported ..."

You need to do Table 6 again so that it presents correlations consistent with APA format.

I think 'three quadrants' should be 'three quarters'?

The paper lacks statistical rigour. Consider on Line 182-184: “An independent t-test was used to test the difference between two means of continuous variables. ANOVA test was used to test the difference between two means of continuous variables.” So it seems a t-test and an ANOVA were both used "to test the difference between two means of continuous variables". There were no hypotheses (planned tests) just research questions looking for significant results. This being the case, should a statistical correction not be used for significance test to control for chance/error?

My comments for the first 128 lines are as follows:

In the abstract, state early that the sample used comprises women only.

Line 48: Consider making ‘aspects’ the end of one sentence and making ‘age’ the first word in a new sentence.

Line 52 and 53. Consider rewriting the sentence “Excessive Facebook use has been proven …” as “Several studies have reported that excessive Facebook use has harmful effects …”

Line 67. Instead of ‘addicts’ use ‘people with additions’ or similar.

Lines 63 & 64. You state “Addictions and emotion regulation have been proven to be linked” yet provide no reference.

 Line 65: It is unclear what “suppression among emotion regulation strategies” means.

Lines 70 & 71: The sentence on these lines does not make sense.

Line 72: This claim “Therefore, psychiatric nurses …” does not follow from the previous sentence.

Line 78: the comma after ‘interest’ has a space before it. Delete it.

Line 79. You mention ‘valid disorder’. You need to explain what you mean by this term, especially ‘valid’.

Line 85 & 86. Consider rewriting “Nearly no researches or little ones have been made in Arab nations” as “Few studies have been conducted in Arab nations …”

Line 86. ‘face book’ should be ‘Facebook’. Also, quotation marks are not needed on Lines 86 & 87.

Lines 87-89. The sentence spanning these three lines is not grammatically correct.

Line: Is the word ‘prospective’ the correct word? It does not make sense.

Line 89: ‘face book’ should be ‘Facebook’.

Line 113. This term ‘standard normal variety’ does make sense.

Line 127. ‘scale items is ranged from’ should be ‘scale items ranged from.’

Line 127. There should be a space between ‘from’ and ‘19’.

Line 128. ‘was’ should be ‘were’.

I realise English may not be your first language, but I struggled to read the paper given the poor English used.

Author Response

Dear Reviewer

I would like to express my sincere appreciation for your thoughtful and constructive comments on our manuscript. Your feedback has been incredibly valuable in helping us to improve the quality of our work and to ensure that it meets the highest standards of academic rigor. We have taken your suggestions into careful consideration as we revise our manuscript for resubmission.

Once again, thank you for your time and attention to our manuscript. We look forward to hearing from you if further modification is needed.

Sincerely

The following section includes the responses for comment:

Statistical analysis section comments done and other editing needed done

Point 1: In the abstract, state early that the sample used comprises women only.

Response 1: modification added in line 19

Point 2: Line 48: Consider making ‘aspects’ the end of one sentence and making ‘age’ the first word in a new sentence.

Response 2:  modification done in line 50

Point 3: Line 52 and 53. Consider rewriting the sentence “Excessive Facebook use has been proven …” as “Several studies have reported that excessive Facebook use has harmful effects …”

Response 3: rewriting the sentence done in line 55-58

Point 4: Line 67. Instead of ‘addicts’, use ‘people with additions’ or similar.

Response 4:  modification done in line 70

Point 5: Lines 63 & 64. You state “Addictions and emotion regulation have been proven to be linked” yet provide no reference.

Response 5:  reference added

 Point 6:Line 65: It is unclear what “suppression among emotion regulation strategies” means.

Response 6: suppression" refers to the intentional inhibition or concealment of emotions, and therefore, it refers to the various techniques people used to manage their emotions.

Point 7: Lines 70 & 71: The sentence on these lines does not make sense.

Response 7:  modification done in line 73-74

Point 8: Line 72: This claim “Therefore, psychiatric nurses …” does not follow from the previous sentence.

Response 8:  modification done

Point 9: Line 78: the comma after ‘interest’ has a space before it. Delete it.

Response 9:  modification done

Point 10: Line 79. You mention ‘valid disorder’. You need to explain what you mean by this term, especially ‘valid’.

Response 10:  modification done in line 82

Point 11: Line 85 & 86. Consider rewriting “Nearly no researches or little ones have been made in Arab nations” as “Few studies have been conducted in Arab nations …”

Response 11: rewriting sentences were done

Point 12: Line 86. ‘face book’ should be ‘Facebook’. Also, quotation marks are not needed on Lines 86 & 87.

Response 12:  modification done

Point 13: Lines 87-89. The sentence spanning these three lines is not grammatically correct.

Response 13:  modification done

Point 14: Line: Is the word ‘prospective’ the correct word? It does not make sense.

Response 14:  modification done in line 93

Point 15: Line 89: ‘face book’ should be ‘Facebook’.

Response 15:  modification done in line 91

Point 16: Line 127. ‘scale items is ranged from’ should be ‘scale items ranged from.’

Response 16:  modification done

Point 17: Line 127. There should be a space between ‘from’ and ‘19’. 

Response 17:  modification done

Point 18: Line 128. ‘was’ should be ‘were’.

Response 18:  modification done in line 133

Reviewer 3 Report

REVIEW: Association between Facebook addiction, depression, and emotional regulation among women

Dear Authors, I was happy to review your paper on an interesting topic.  I see it worth publishing as a research article after some basically structural revisions and small detail additions. I suggest that you keep your Discussion section as a starting point when making your revisions.

Traditionally, the main content of this section is to elaborate how your results relate to earlier ones, what is their theoretical and practical significance, and how they guide towards further research. In this version, a main part of the earlier research is presented in Discussion. It should be presented in Introduction and give arguments in favor of your research when you point out the existing knowledge gap which you are going to fill.

Another point is that the first explanations of your results are in Discussion, when they should be in Results -section. I appreciate your way you have written introductions to your tables that are also clear and easy to read. I suggest that after every table you write your first explanations derived from them. Any further interpretations and comparisons with earlier results find their proper place in Discussion. Remove and rewrite, do not repeat.

Some details:

While concentrating on the lack of research on Arab women and Facebook, you neglect to inform the reader about the situation of Arab men or the whole population. Are there research results available?

The reader needs to know more about Dakahlia, the exact sites of research, and the living and working conditions of women in those areas, preferably already in the beginning of the research report. The roles of Social Clubs remain unclear for readers, who do not know the culture. Further questions: How exactly were the participants invited or found? Did the participants get to know each other via WhatsApp -group or was their anonymity maintained? Why is the number of children of the participants not asked, as taking care of children might lessen the time related possibilities to get addicted to Face Book, but rather produce strain in conciliating work and family, and thus making women more vulnerable in issues of emotional regulation.

My last comment: I suggest that you make a fresh evaluation of the limitations of your research by giving less emphasis on generalizability in whole or male populations but more on the social and cultural context. Without relevant knowledge, it seems to me that the social and cultural context of this study protects your participants against Face Books addiction and the other psychological problems considered in your framework. The issue of generalizability issue is partly solved when stating clearly that this is a study about women in Arab rural communities, also by adding this to the title of the paper.

All the best for your revision work, sincerely

Your Reviewer   

Author Response

Dear Reviewer,

Thank you for taking the time to review our manuscript. We appreciate your comments and suggestions for improving the article. We have carefully considered your feedback and made the modifications to address your concerns in the revised manuscript.

We hope that our revisions have addressed your concerns and that you will consider our manuscript for publication. If you have any further suggestions or modifications, please does not hesitate to let us know.

Sincerely,

Round 2

Reviewer 2 Report

I note the sentence "More over this result could be related to busy of the women" This does not make sense. Fix it.

Fine

Author Response

Dear Reviewer

Thank you for your meticulous review and for providing me with such valuable feedback. Your insights have been instrumental in improving the quality of my work. I am grateful for your expertise in this field and for the time and effort you have dedicated to reviewing my paper. I hope that the modifications I made will address your insights and further enhance the quality of my research.
Sincerely